# Ramon Flour (*Brosimum alicastrum* Swartz) Ameliorates Hepatic Lipid Accumulation, Induction of AMPK Phosphorylation, and Expression of the Hepatic Antioxidant System in a High-Fat-Diet-Induced Obesity Mouse Model

**DOI:** 10.3390/antiox12111957

**Published:** 2023-11-02

**Authors:** Trinidad Eugenia Cu-Cañetas, Laura A. Velázquez-Villegas, Mariana Manzanilla-Franco, Teresa del Rosario Ayora-Talavera, Juan José Acevedo-Fernández, Enrique Barbosa-Martín, Claudia C. Márquez-Mota, Adriana M. López-Barradas, Lilia G. Noriega, Martha Guevara-Cruz, Ana Ligia Gutiérrez-Solís, Azalia Avila-Nava

**Affiliations:** 1Escuela de Salud, Universidad Modelo, Mérida 97130, Yucatán, Mexico; trinidadcu8@gmail.com (T.E.C.-C.); mariana.mf4@gmail.com (M.M.-F.); enrique.barbosa89@hotmail.com (E.B.-M.); 2Departamento de Fisiología de la Nutrición, Instituto Nacional de Ciencias Médicas y Nutrición Salvador Zubirán (INCMNSZ), Ciudad de México 14080, Mexico; laus0505@gmail.com (L.A.V.-V.); adrimar24@gmail.com (A.M.L.-B.); lgnoriegal@gmail.com (L.G.N.); marthaguevara8@yahoo.com.mx (M.G.-C.); 3Centro de Investigación y Asistencia en Tecnología y Diseño del Estado de Jalisco (CIATEJ) A.C., Subsede Sureste, Mérida 97302, Yucatán, Mexico; tayora@ciatej.mx; 4Departamento de Fisiología y Fisiopatología, Facultad de Medicina, Universidad Autónoma del Estado de Morelos (UAEM), Cuernavaca 62350, Morelos, Mexico; juan.acevedo@uaem.mx; 5Departamento de Nutrición Animal y Bioquímica, Facultad de Medicina Veterinaria y Zootecnia, Universidad Nacional Autónoma de México (FMVZ-UNAM), Ciudad de México 04510, Mexico; c.marquez@unam.mx; 6Hospital Regional de Alta Especialidad de la Península de Yucatán (HRAEPY), Mérida 97130, Yucatán, Mexico; ganaligia@gmail.com

**Keywords:** Ramon flour, lipid metabolism, adipocytes, fatty liver, polyphenols

## Abstract

Excessive consumption of fat and carbohydrates, together with a decrease in traditional food intake, has been related to obesity and the development of metabolic alterations. Ramon seed is a traditional Mayan food used to obtain Ramon flour (RF) with high biological value in terms of protein, fiber, micronutrients, and bioactive compounds such as polyphenols. However, few studies have evaluated the beneficial effects of RF. Thus, we aimed to determine the metabolic effects of RF consumption on a high-fat-diet-induced obesity mouse model. We divided male *BALB/c* mice into four groups (*n* = 5 each group) and fed them for 90 days with the following diets: Control (C): control diet (AIN-93), C + RF: control diet adjusted with 25% RF, HFD: high-fat diet + 5% sugar in water, and HFD + RF: high-fat diet adjusted with 25% RF + 5% sugar in water. The RF prevented the increase in serum total cholesterol (TC) and alanine transaminase (ALT) that occurred in the C and HFD groups. Notably, RF together with HFD increased serum polyphenols and antioxidant activity, and it promoted a decrease in the adipocyte size in white adipose tissue, along with lower hepatic lipid accumulation than in the HFD group. In the liver, the HFD + RF group showed an increase in the expression of β-oxidation-related genes, and downregulation of the fatty acid synthase (*Fas*) gene compared with the HFD group. Moreover, the HFD + RF group had increased hepatic phosphorylation of AMP-activated protein kinase (AMPK), along with increased nuclear factor erythroid 2-related factor 2 (NRF2) and superoxide dismutase 2 (SOD2) protein expression compared with the HFD group. Thus, RF may be used as a nutritional strategy to decrease metabolic alterations during obesity.

## 1. Introduction

Obesity has become a public health problem due to its increasing prevalence worldwide. The World Obesity Federation estimated in 2020 that approximately 770 million adults globally had obesity [1]. Obesity is defined by the World Health Organization as an abnormal or excessive accumulation of fat that can be harmful to health [2]. This pathology has been related to the development of many biochemical abnormalities in lipid and glucose metabolism, which are associated with complications such as glucose intolerance and dyslipidemia [3]. Under these conditions, there is increased mobilization and release of lipids, which promotes their accumulation in non-specific tissues such as the liver. This process is mediated by AMP-activated protein kinase (AMPK) and oxidative stress. Disruption of AMPK activation, in turn, decreases fatty acid synthesis (lipogenesis) and inhibits fatty acid oxidation, causing hepatic lipid accumulation. AMPK is a sensor of the cellular energy status; thus, when it is inactive, there are alterations in energy production and mitochondrial homeostasis [4]. Mitochondrial dysfunction generates oxidative stress by increasing the production of reactive oxygen species and depleting the antioxidant system, which also enhances dysregulation in lipid metabolism in the liver [5]. 

These metabolic complications have been associated with an acculturation process, which often involves an increase in the consumption of fat and carbohydrates in the diet and a decrease in the intake of traditional foods. On the other hand, emerging data indicate that foods that connect ethnicity with dietary intake patterns may promote healthy lifestyles [6]. Traditional foods have been used for many years due to the beneficial effects attributed to their consumption, in addition to their cultural and heritage contributions [7]. Currently, there is a trend to promote the consumption of traditional foods due to their beneficial effects, which could potentially improve health. However, knowledge of the benefits is often based on observational studies and physiological arguments [8,9,10]. Among traditional Mexican foods is the flour obtained from Ramon seed, which is known as Maya nut. It has been used since pre-Hispanic times, being known to have been involved in food preparation within the ancient Maya civilization. Throughout history, Ramon flour (RF) has been an important food source in periods of scarcity of the usual crops due to its chemical composition [11,12]. RF is a source of carbohydrates (81.5%), protein (12.1%), fat (1.7%), crude fiber (6.6%), and minerals (4.7%), and it has also been reported to provide bioactive compounds including vitamins and polyphenols [13]. Polyphenols are bioactive compounds that can modulate different metabolic pathways through signal transduction, which, in turn, activate lipid and glucose metabolism, the cellular redox state, and antioxidant defenses [14]. Thus, based on its chemical composition, RF could be used as a nutritional strategy against metabolic diseases and to generate new food products with improved nutritional contents. A previous study showed that the addition of RF in the preparation of tortillas increased the amount of fiber and minerals compared with tortillas made with wheat flour [15]. Another study showed that substitution of wheat flour with RF in a cookie recipe increased the protein and fiber contents [13]. Despite these developments and the knowledge that RF could be a good source of bioactive compounds that could potentially improve health, there have been only a few studies that show the effects of RF on pathological conditions. Thus, we aimed to determine the metabolic effects of RF consumption on a high-fat-diet-induced obesity mouse model.

## 2. Materials and Methods

### 2.1. Animal Experiments

Twenty male *BALB/c* mice (32 g) were obtained from the Dr. Hideyo Noguchi Regional Research Center. The animals were maintained in individual cages at room temperature, with a 12 h photoperiod, and with free access to water and food. The procedures were approved by the Institutional Animal Care and Use Committee of the Faculty of Medicine of the Universidad Autónoma del Estado de Morelos (approval number 005/2021).

### 2.2. Study Design and Diets

The animals were divided into the following groups (*n* = 5 each group): Control (C): control diet (AIN-93); C + RF: control diet adjusted with 25% Ramon flour; HFD: high-fat diet + 5% sugar in water; and HFD + RF: high-fat diet adjusted with 25% Ramon flour + 5% sugar in water. The body weight and food consumption were measured every 7 days during the experiment. The water consumption every 2 days was also considered in the HFD and HFD + RF groups. The control diet was prepared based on the requirements of the American Institute of Nutrition formulation (AIN-93) [16], and the HFD diet was prepared with modifications in the lipid/carbohydrate content reported previously [17]. The diets that contained RF were adjusted at 25% to maintain the nutritional requirements of all macronutrients (Appendix A). Blood was collected from the vena cava, centrifuged (1500× *g* for 10 min), and the serum was transferred to another tube and stored at −70 °C until analysis. Liver and adipose tissue were collected and stored at –70 °C. Liver and adipose tissue samples were fixed in formalin to perform hematoxylin and eosin staining.

### 2.3. Chemical Analysis of Experimental Diet

The carbohydrate, protein, lipid, fiber, and mineral composition of the diet was analyzed based on the techniques recommended by the Association of Official Analytical Chemists (AOAC) International [18]. 

### 2.4. Evaluation of the Total Polyphenol Content

The total polyphenol content in RF and serum was measurement using the Folin–Ciocalteu method with modifications [19]. The total polyphenol content is expressed as milligrams of gallic acid equivalents (GAE/dL or GAE/g).

### 2.5. Determination of the Antioxidant Activity

The antioxidant activity in serum was determined using a florescence method, namely oxygen radical absorbance capacity (ORAC) [20]. This assay evaluated the fluorescence signal at 485 nm (excitation) and 535 nm (emission) for 90 min at 1 min intervals. Finally, the area under curve (AUC) was calculated via point-to-point integration of the fluorescence signal. The results are expressed as micromoles of Trolox equivalents per milliliter (Trolox equivalents, μmol/mL).

### 2.6. Oral Glucose Tolerance Test

An oral glucose tolerance test (OGTT) was performed at the end of the study. Mice that had been deprived of food for 8 h were administered an oral glucose load (2 g/kg body weight). At 0, 15, 30, 60, 90, and 120 min after administration of the glucose load, the blood glucose level was measured with a portable blood glucose meter (ACCU-CHEK, Roche Diagnostics, Indianapolis, IN, USA). 

### 2.7. Serum Biochemical Parameters and Histological Analyses

Serum biochemical parameters such as total cholesterol (TC), triglycerides, glucose, and alanine transaminase (ALT) were analyzed in a COBAS C111 device (Roche, Basel, Switzerland). Histological characteristics of liver and white adipose tissue (WAT) were evaluated using a Leica Qwin image-analyzer system on a Leica DMLS microscope (Leica DM750 Wetzlar, Germany). Tissues were fixed in 4% paraformaldehyde and then sectioned; 5 μm thick sections were stained with hematoxylin and eosin. The adipocyte area of WAT was calculated with Adiposoft software version 1.16 (National Institutes of Health, Bethesda, MD, USA).

### 2.8. Evaluation of Gene Expression Using Quantitative Real-Time Polymerase Chain Reaction

Total RNA was extracted from liver samples with the TRIzol reagent ((Invitrogen, Carlsbad, CA, USA), following the manufacturer’s instructions. The abundance of messenger RNA (mRNA) was determined using SYBER GREEN assays via real-time quantitative polymerase chain reaction (PCR). Cyclophilin was use as a reference for normalization. The primers are listed in Appendix A. 

### 2.9. Quantification of Protein Abundance Using Western Blotting Analysis

Proteins from the liver were extracted with the used of ice-cold radioimmunoprecipitation assay (RIPA) buffer, which contained complete mini protease inhibitor (Roche Diagnostics). The Qubit Protein Assay kit and Qubit 3.0 equipment were used to quantify proteins according to the manufacturer’s instructions. Aliquots from each sample were stored at −70 °C until use. Protein (25 μg) was separated using sodium dodecyl sulfate–polyacrylamide gels (8% and 12%) and transferred onto polyvinylidene difluoride (PVDF) membranes. These were incubated with 5% nonfat dry milk (blocking solution) for 1 h to block nonspecific protein binding. Then, the membranes were incubated in blocking solution overnight at 4 °C with one of the following primary antibodies: AMPK1/2 (1:1000; sc-, Santa Cruz Biotechnologies, Santa Cruz, CA, USA), P-AMPK (Thr-172) (1:1000; sc-, Santa Cruz Biotechnologies, Santa Cruz, CA, USA), NRF2 (1:500; #12721 Cell Signaling, Danvers, MA, USA), SOD2 (1:3000; 06-984 Merk Millipore Burlington, MA, USA), or HSP70 (1:10,000; SAB4200714 Sigma Aldrich, San Luis, MO, USA). Subsequently, the membranes were incubated for 1 h at room temperature with an anti-rabbit (1:40,000; ab6789, Abcam, Cambridge, UK) or anti-mouse (1:40,000; ab6751, Abcam) horseradish peroxidase-conjugated secondary antibody. A chemiluminescent detection reagent (Merk Millipore) and the ChemiDoc™ XRS and System Image Lab™ Software version 6.1 (Bio-Rad, Hercules, CA, USA) was used for visualization. Glyceraldehyde 3-phosphate dehydrogenase (GAPDH) protein was used as the loading control; it was detected with a primary antibody from Abcam (1:40,000; ab). The ImageJ software version 1.5.3 (National Institutes of Health, Bethesda, MD, USA) was used for densitometric analysis of the immunoblot bands. Western blotting was performed using three independent blots for each protein of interest.

### 2.10. Statistical Analysis

The data are shown as the mean ± standard error of mean (SEM). One-way analysis of variance (ANOVA) followed by Tukey’s *post hoc* test was used for statistical comparisons (GraphPad Prism 7.0, GraphPad Software, San Diego, CA, USA). Differences between the experimental groups are shown with letters, where a > b > c > d. A *p*-value < 0.05 was considered to be statistically significant.

## 3. Results

### 3.1. Addition of Ramon Flour to the Control or High-Fat Diet Did Not Change Their Chemical Composition

There were no differences between the C and C + RF diets or between the HFD and HFD + RF diets in the carbohydrate, protein, or lipid contents (Table 1). This corroboration was important to ensure we had correctly adjusted the diets to maintain the same consumption of macronutrients among the groups. 

### 3.2. Ramon Flour Consumption Did Not Change Body Weight Gain, Food, or Energy Intake

We did not observe significant differences in body weight between the groups during the study or in the final weight gained (Figure 1A,B). The mean food consumption of the HFD group was lower compared with the C group (5.98 ± 0.43 vs. 3.81 ± 0.24 g/day, *p* < 0.01), while the HFD + RF group had higher food intake compared with the C + RF group (5.91 ± 0.34 vs. 4.07 ± 0.41 g/d, *p* > 0.05) (Figure 1C,D). However, when comparing the energy intake (kcal/day), there were no differences between the groups (Figure 1E,F). The lack of a difference is because the reported energy intake considers the calories consumed from the sugar water. This approach ensured that the consumption between the groups was not different and that any effects we observed could be due to the addition of RF.

### 3.3. Ramon Flour Prevented an Increase in Total Cholesterol and Alanine Aminotransferase upon High-Fat Diet Feeding

Consumption of an HFD is associated with alterations in serum metabolic parameters, so we analyzed the levels of some of these biomarkers. There were no differences in the glucose and triglyceride concentrations between the groups (Figure 2A,B). However, the HFD group presented a higher TC level compared with the C group (228 ± 5.44 vs. 178 ± 19.5 mg/dL, *p* < 0.05) (Figure 2C). Interestingly, the HFD + RF group did not present the increase in the TC level observed in the HFD group (191 ± 5.14 vs. 228 ± 5.44 mg/dL, *p* < 0.05). Moreover, the C + RF group showed a lower TC level than the C group (Figure 2C). 

We wanted to analyze a biomarker that reflects hepatic damage; for that purpose, we determined the level of ALT. As expected, the HFD group showed the highest ALT level (31.4 ± 4.59 mg/dL, *p* < 0.01). Interestingly, the HFD + RF group showed a decrease in ALT compared with the HFD group (11.4 ± 2.41 mg/dL, *p* < 0.01) (Figure 2D), suggesting that RF consumption could prevent liver damage upon HFD feeding.

### 3.4. Ramon Flour Decreased Glucose Levels in the Oral Glucose Tolerance Test upon High-Fat Diet Feeding

Alterations in glucose metabolism usually occur after the consumption of a chronic excess of lipids and carbohydrates. To analyze the effect of RF on glucose metabolism, we performed an OGTT. At 90 and 120 min of the OGTT, the glucose levels of the HFD + RF group were significantly lower compared with the HFD group (*p* < 0.05), suggesting that the consumption of RF alongside the HFD improves glucose tolerance 2 h after oral glucose administration. However, the AUC during the OGTT did not show differences between the groups (Figure 3B). 

### 3.5. Ramon Flour Consumption Increased the Serum Antioxidant Activity and Polyphenol Content

To evaluate the potential effect of the bioactive compounds found in RF, we evaluated the serum antioxidant activity and polyphenol content. The HFD group had significantly decreased antioxidant activity (635 ± 146 Trolox equivalents (μmol/mL), *p* < 0.05) compared with the other groups (Figure 4A). Interestingly, the addition of RF to the HFD rescued the serum antioxidant activity compared with the HFD group (1989 ± 263 Trolox equivalents (μmol/mL), *p* < 0.05), reaching a level similar to the C group (Figure 4A). These results are consistent with the polyphenol content, which was higher in the HFD + RF group compared with the HFD group (11.98 ± 0.28 vs. 9.35 ± 0.28 GAE/mL, *p* < 0.05) (Figure 4B). This difference may be associated with the total polyphenol content of RF (52.9 ± 3.78 GAE/g). 

### 3.6. Ramon Flour Consumption Decreased Lipid Accumulation and Adipocyte Size in White Adipose Tissue

Excess lipid accumulation in adipocytes has been associated with metabolic alterations in WAT. To evaluate the effect of RF on WAT, we performed histological analysis and measured the adipocyte area in the experimental groups. There were no differences in adipocyte morphology between the C and C + RF groups (Figure 5A,B). There was an increase in the adipocyte size in the HFD group (Figure 5C), indicating adipose tissue hypertrophy; this increase was prevented in the HDF + RF group (Figure 5D). Quantification of the adipocyte area confirmed a significant increase in adipocyte size (Figure 5E) and the frequency of larger adipocytes (Figure 5F) in WAT in the HFD group compared with the C group (813 ± 31.1 vs. 436 ± 6.06 mm^2^, *p* < 0.001) as well as the HFD + RF group (498 ± 35.6 vs. 813 ± 31.1 mm^2^, *p* < 0.001). These data indicate that the consumption of RF prevented WAT hypertrophy after HFD feeding.

### 3.7. Ramon Flour Consumption Prevented Hepatic Lipid Accumulation

We performed macroscopic and microscopic analyses of the liver morphology. Macroscopically, the livers from the C and C + RF groups had a normal glossy appearance (Figure 6A,B, respectively). In contrast, the livers from the HFD group were enlarged and had a yellow/pale appearance (Figure 6C), suggesting that lipids had accumulated. Interestingly, the livers from the HFD + RF group showed a similar appearance to the C group (Figure 6D). Microscopic analysis of liver tissue showed a normal structure in the C (Figure 6E) and C + RF (Figure 6F) groups. Meanwhile, the HFD group (Figure 6G) showed the presence of multiple lipid droplets, suggesting lipid accumulation in this tissue. In contrast, the liver histology of the HFD + RF group (Figure 6H) was very similar to the C group. Based on these macroscopic and microscopic analyses of the liver morphology, we hypothesize that RF plays a role in lipid metabolism, thus preventing lipid accumulation in the liver even with consumption of the HFD. 

### 3.8. Ramon Flour Induced the Hepatic Expression of Genes Involved in β-Oxidation and Prevented the Induction of the Lipogenic Gene Fatty Acid Synthase upon High-Fat Diet Feeding

Regulation of hepatic lipid accumulation is associated with changes in the expression of genes involved in lipid metabolism. The C + RF group showed a significant increase in the expression of the carnitine palmitoyl transferase 1 (*Cpt1*) gene compared with the C group (Figure 6I). Interestingly, the HFD + RF group also had a significant increase (2.6-fold change) in the *Cpt1* gene (Figure 6I), suggesting that the consumption of RF could increase β-oxidation in the liver. Interestingly, the peroxisomal β-oxidation gene acyl-CoA oxidase 1 (*Acox1*) was also upregulated in the C + RF group, and there was a similar trend in the HFD + RF group (Figure 6J), suggesting that RF may also induce β-oxidation in peroxisomes. On the other hand, the HFD group had a 2.5-fold increase in the expression of fatty acid synthase (*Fas*), a gene involved in lipogenesis (Figure 6K). Changes in mitochondrial oxidation led to the modulation of genes associated with the redox status, such as superoxide dismutase 2 (*Sod2*), which encodes a mitochondrial protein regulated by oxidative stress. In this sense, RF intake alongside the C diet or the HFD significantly increased the expression of the *Sod2* (Figure 6L) gene by 2.2-fold compared with the groups without RF. These results suggest that RF could induce β-oxidation, and the expression of antioxidant genes could be activated in response to this increase to avoid oxidative damage associated with an increase in reactive oxygen species production.

### 3.9. Ramon Flour Consumption Promoted the Activation of AMP-Activated Protein Kinase and Increased the Abundance of Antioxidant Proteins in the Liver

To corroborate the potential effect of RF on modulating hepatic lipid metabolism, we evaluated the activation of the energetic sensor AMPK via Western blotting. Interestingly, the HFD + RF group showed a 1.5-fold increase in P-AMPK compared with the HFD group (*p* < 0.05) (Figure 7A). We observed the same trend in the C + RF group compared with the C group (*p* < 0.05) (Figure 7A), suggesting that RF may stimulate the expression of fatty acid oxidation and inhibit the expression of lipogenic genes via AMPK. 

Because we observed an upregulation of genes involved in the response to maintain the redox state, we wanted to corroborate this effect by measuring the abundance of proteins involved in this process, including HSP70, NRF2, and SOD2. The HFD group showed a significantly increased HSP70 abundance compared with the C group (Figure 7B), indicating that the consumption of an overload of lipids induces a stress response in the liver. Interestingly, the HFD + RF group had a 2.7-fold increase in NRF2 abundance compared with the HFD group (*p* < 0.001) (Figure 7C). This effect was associated with a 1.7-fold increase in the abundance of SOD2 in the same experimental group (*p* < 0.001), which confirms that RF consumption induces the expression of antioxidant proteins, possibly to protect hepatocytes from oxidative stress known to be induced by HFD feeding. Notably, the C + RF group did not show increased NRF2 and SOD2 abundance compared with the C group (Figure 7C,D). This result suggests that the effect of RF is only needed upon HFD feeding, a situation in which there is an excess of lipids that are directed to the mitochondria for their oxidation to avoid hepatic lipid accumulation and oxidative stress.

## 4. Discussion

The results of present work showed that the addition of RF to the HFD decreased lipid accumulation and AMPK activation in the liver and upregulated the expression of genes involved in β-oxidation and the antioxidant system in an HFD-induced obesity mouse model. To our knowledge, this is the first evidence of the beneficial effects of RF consumption at the molecular level.

The actual changes in eating patterns, characterized by excessive consumption of lipids and refined carbohydrates and a decrease in traditional or local foods, are related to the development of metabolic alterations and other pathologies [21]. One of the main metabolic alterations observed upon HFD feeding and dysfunction in energy homeostasis is the development of a fatty liver, which is characterized by microvascular or macrovascular steatosis [22,23]. Hepatic lipid deposition is mainly derived from fatty acids coming from the lipolysis of adipocytes and dietary lipids. In fact, the excessive consumption of lipids and sugars generates an imbalance, which, in turn, alters the absorption, synthesis, oxidation, and export of lipids. Dysregulation of these processes is associated with the development of other complications such as glucose intolerance, dyslipidemia, and cardiovascular disease [24]. Thus, it is important to search for nutritional strategies that may help to prevent or reduce these metabolic alterations. In line with this view, resuming the consumption of RF, a traditional food, could be a measure to consider, with potential health and sustainability benefits [15]. In fact, several communities have included RF as part of their daily diet. Therefore, the U.S. Food and Drug Administration has granted it a certificate as a Generally Recognized as Safe food [25]. With that said, although the consumption of RF has been widely reported in these communities, there is little evidence regarding its effects on improving health. 

Considering the potential benefits that RF consumption could generate, we adjusted the chemical composition of the diets in this study. The adjustment of diet macronutrients allows us to suggest that the observed changes are associated with the specific presence of RF. We observed that the addition of RF to the HFD promoted a decrease in the TC level compared with the HFD group. Similar results were recently reported in elderly individuals, where the ingestion of a beverage with RF significantly decreased low-density lipoprotein cholesterol (14.8%) and urea (33.1%) and increased total polyphenols (7.8%) in circulation [26]. Moreover, we observed an increase in the circulating polyphenol content and antioxidant activity in the HFD + RF group. These beneficial effects are associated with the chemical composition of RF, as it contains compounds with nutritional properties, including soluble and insoluble fiber, vegetal protein, and polyphenols [15]. In fact, RF present in the diets showed a similar total polyphenol content as in a previous report [15]. Only a few studies have reported the polyphenol content and identified specific polyphenols and their derivatives in Ramon seed. One of these studies showed the presence of bioactive compounds including hydroxycinnamic, gallic, vanillic, caffeic, and coumaric acids [27]. Meanwhile, another study identified chlorogenic, caffeoylquinic, and vanillic acids in these seeds [15]. Hydroxycinnamic acid and its intermediate derivative products such as caffeic acid, *p*-coumaric acid, chlorogenic acid, curcumin, and hydroxycinnamic acids are abundant in fruits, vegetables, cereals, and the seeds of several fruits [28,29]. Interestingly, there is evidence that most of the hydroxycinnamic acid derivatives can act against lipid deposition and increase hepatic lipid metabolism [30]. The proposed molecular mechanism by which these types of compounds lower cholesterol levels is through regulation of the expression of the *CYP7A1* gene, which encodes the rate-limiting enzyme for the synthesis of bile salts and is part of one of the regulatory pathways of cholesterol metabolism [31]. In fact, studies have shown that foods containing polyphenols, such as tea and curcumin, have a hypocholesterolemic effect [32,33]. In addition, a study showed that administration of ferulic acid decreased hepatic lipid deposition, including triglycerides and TC, in HFD-fed mice [34]. Similarly, consumption of caffeic acid and chlorogenic acid significantly inhibited FAS activity, while they increased fatty acid β-oxidation activity in mouse liver compared with the high-fat group [35]. These bioactive compounds present in Ramon seeds could activate intracellular signaling pathways and induce transcription factors, which, in turn, regulate the expression of genes and proteins involved in lipid and carbohydrate metabolism [36]. A molecular mechanism that could explain this metabolic regulation by polyphenols is based on activation of peroxisome proliferator-activated receptor (PPAR) α, which is a nuclear factor involved in the modulation of gene expression of β-oxidation enzymes. In the liver, PPARα regulates lipid metabolism, controls nutrient and energy homeostasis, and prevents the development of abnormalities that may lead to hepatic lipid accumulation [37]. However, this mechanism is not the only one through which polyphenols can affect metabolism; these compounds can also activate AMPK, which is an energy sensor that maintains cellular energy homeostasis [38]. There is evidence of decreased activation of AMPK by phosphorylation during obesity, which is accompanied by oxidative stress and inflammation [39,40].

In this sense, the present results suggest that the polyphenols found in RF induce AMPK phosphorylation [41]. Therefore, activation of AMPK by polyphenols contained in RF could explain the beneficial effects observed in the liver of the HFD + RF group. AMPK phosphorylation at Thr172 through upstream kinases decreases malonyl-CoA levels, which, in turn, activates CPT1, leading to the increased transport of acyl-CoA to mitochondria and enhanced β-oxidation [42]. This could explain the effects of RF in the liver given the decrease in lipid vacuoles in hepatic tissue together with reduced *Fas* expression and upregulated *Cpt1* expression. In fact, modulation of these metabolic enzymes has been associated with physiological activity or dysfunction of lipid metabolism, where β-oxidation is mainly upregulated by CPT1, while FAS is a major regulator of fatty acid lipogenesis and triglyceride synthesis [43]. The effect of RF in modulating the expression of lipid metabolism genes has also been observed for chia flour. In hamsters, compared with the HFD group, the HFD + chia flour group showed a 46% reduction in hepatic lipid droplets and gene expression changes: a decrease in sterol regulatory element binding transcription factor 1 (*Srebf1*) and an increase in *Cpt1* [44]. Interestingly, we noted beneficial effects in the C + RF group: significantly increased *Acox1* and *Cpt1* gene expression and induction of AMPK phosphorylation, suggesting that RF increases β-oxidation even without an overload of lipids. 

The effects of RF in the HFD group are particularly interesting because HFD feeding is known to generate a low-grade inflammatory state. Under these conditions, there is an increase in pro-inflammatory interleukins and reactive oxygen species that will eventually generate cellular and oxidative stresses [45]. However, these alterations can be mediated by HSPs and the endogenous antioxidant defense. Of note, HDP70 protein abundance was not difference between the HFD and HFD + RF groups. This outcome could be explained by the fact that the elevation of HSP70 is associated with more advanced fatty liver damage, such as hepatic steatosis [46].

One of the initial alterations in the development of a fatty liver is oxidative stress, which also plays a fundamental role in metabolic disorders [47]. In fact, there is evidence that the progression to more advanced liver damage is due to prolonged exposure to oxidative stress, which is an imbalance between reactive oxygen species and antioxidants. This is exacerbated in pathological conditions such as obesity, where there is a decrease in the endogenous antioxidant response [5]. Under healthy conditions, activation of the antioxidant response is not always necessary to maintain or to recover the redox status. However, under stress conditions such as HFD feeding, antioxidant proteins should be induced. The antioxidant response is induced through the transcription factor NRF2, which translocate from the cytosol to the nucleus and binds to the sequence called the antioxidant response element to induce the expression of antioxidant genes such as SOD2 [48]. Nevertheless, the evidence shows that there is an NRF2 deficiency in animal models fed with HFD, leading to metabolic stress and development of steatohepatitis [49]. Our results agree with this evidence: We observed similar NRF2 protein abundance in the HFD and C groups, suggesting that the HFD group will most likely be unable to cope properly with the stress response. On the other hand, the addition of RF to the HFD increased NRF2 abundance, suggesting that RF helps to properly induce the antioxidant response upon HFD feeding. The antioxidant activity of polyphenols present in RF can be direct or indirect. The direct mechanism is by donating electrons to reactive oxygen species and neutralizing them. Indirectly, polyphenols modify the interaction between NRF2 and Keap1, which is the negative regulator of this protein. Once NRF2 is free, it can translocate to the nucleus and induce the expression of antioxidant genes [50].

It is known that activation of NRF2 prevents metabolic alterations, reducing adipose tissue inflammation and insulin resistance [51,52,53]. Previous studies have demonstrated that bioactive compounds found in traditional foods increase the abundance of antioxidant enzymes. Feeding mice chia oil alongside an HFD ameliorated oxidative damage in the liver due to an increase in NRF2 and SOD protein abundance [54]. Similarly, consumption of Bee bread, which is formed from the fermentation of nectar, pollen, and digestive enzymes secreted from the bee’s salivary glands, activated the Keap1/NRF2 pathway, reduced oxidative stress and improved lipid metabolism in the liver of obese rats [55]. Although the present results showed the beneficial effects of RF in a mouse model of diet-induced obesity, additional studies are required to elucidate the mechanism of action through which all of these metabolic regulations are carried out. Therefore, the addition of antioxidant compounds, such as polyphenols, to the diet represents a strategy to prevent or reduce oxidative stress and thus to reduce the complications associated with it. 

The present study has some limitations. First, we observed no significant difference in the body weight of the animals among the different groups. Nevertheless, we observed metabolic alterations associated with HFD consumption. Another limitation is the absence of specific identification of bioactive compounds. As such, the benefits of RF may be attributed to the combination of various compounds, including fiber, vegetal protein, and unidentified bioactive compounds. Although we determined the total polyphenol content in serum after consumption of the diets, we did not identify specific compounds. Moreover, both the control diet and the HFD had a higher fiber content. Strikingly, despite this higher fiber content, we did not observe the associated benefits in these groups. It is worth noting that we demonstrated the benefits of RF at both the physiological and biochemical levels, particularly in the liver at the molecular level.

In summary, the present results demonstrate that RF prevents biochemical alterations induced by obesity. Thus, consumption of traditional foods, such as RF, can be used as a low-cost and easily accessible nutritional strategy for the prevention and treatment of metabolic diseases.

## 5. Conclusions

Our results suggest that consumption of RF increases circulating antioxidant activity and ameliorates biochemical alterations, lipid accumulation in the liver, and hypertrophy of WAT related to obesity. Furthermore, this scientific evidence supports the beneficial effects of RF as a part of dietary strategies.

## Figures and Tables

**Figure 1 antioxidants-12-01957-f001:**
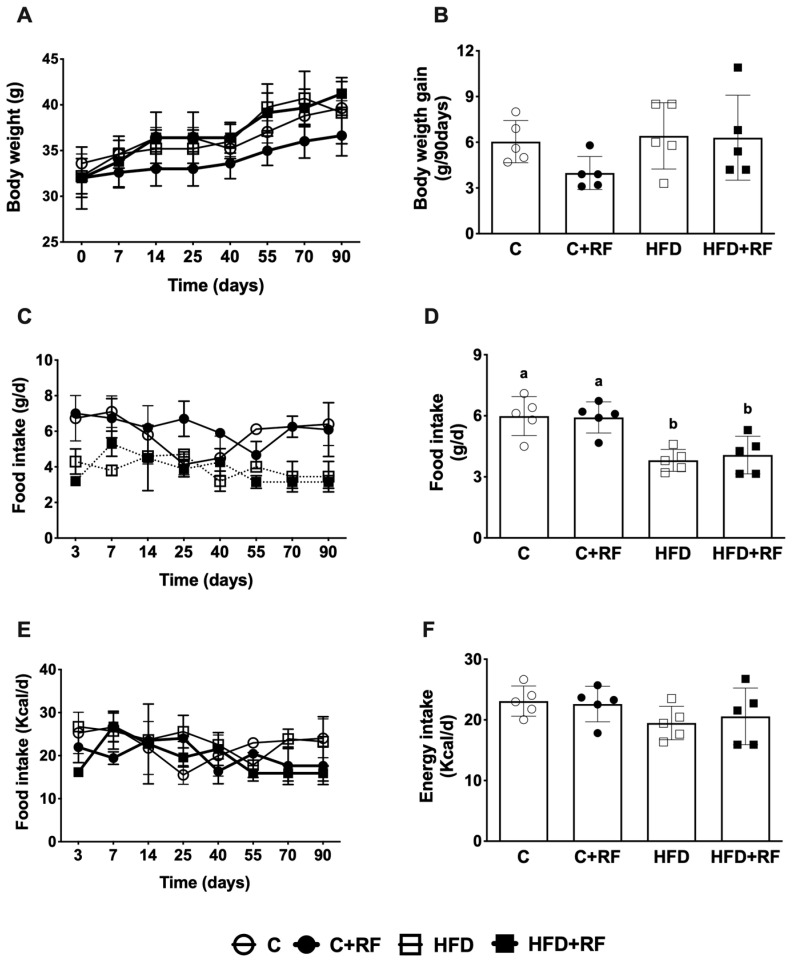
Effect of Ramon flour on a high-fat-diet-induced obesity mouse model, on body weight (**A**), mean body weight gain (**B**), food consumption (**C**), mean food consumption (**D**), energy intake (**E**), and mean energy intake (**F**). Data are expressed as the mean ± SEM (*n* = 5). The data were analyzed using a one-way analysis of variance (ANOVA) test with a *post hoc* Tukey’s test. Differences between experimental groups are shown with letters, where a > b. Statistical differences are considered with a value of *p* < 0.05. Control diet (AIN-93); C + RF: control diet adjusted with 25% Ramon flour; HFD: high-fat diet + 5% sugar in water; and HFD + RF: high-fat diet adjusted with 25% Ramon flour + 5% sugar in water.

**Figure 2 antioxidants-12-01957-f002:**
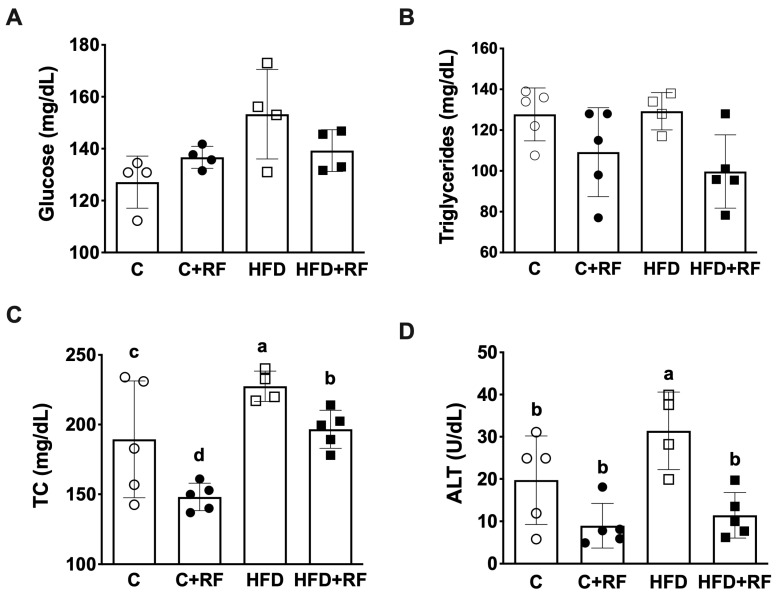
Biochemical parameters in serum after consumption of Ramon flour on a high-fat-diet-induced obesity mouse model; glucose (**A**), triglycerides (**B**), TC (**C**), and ALT (**D**). Data are expressed as the mean ± SEM (*n* = 5). The data were analyzed using the one-way analysis of variance (ANOVA) test with a *post hoc* Tukey’s test. Differences between experimental groups are shown with letters, where a > b > c > d. Statistical differences are considered with a value of *p* < 0.05. C: control diet (AIN-93); C + RF: control diet adjusted with 25% Ramon flour; HFD: high-fat diet + 5% sugar in water; and HFD + RF: high-fat diet adjusted with 25% Ramon flour + 5% sugar in water; TC: total cholesterol; ALT: alanine transaminase.

**Figure 3 antioxidants-12-01957-f003:**
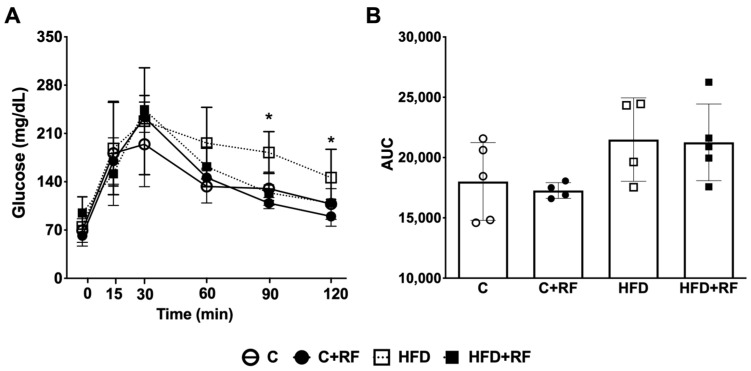
Oral glucose tolerance test (**A**) and area under the curve (**B**) after the consumption of Ramon flour. Data are expressed as the mean ± SEM (*n* = 5). The data were analyzed using a one-way analysis of variance (ANOVA) test with a *post hoc* Tukey’s test. * Differences with respect to HFD group. Statistical differences are considered with a value of *p* < 0.05. C: control diet (AIN-93); C + RF: control diet adjusted with 25% Ramon flour; HFD: high-fat diet + 5% sugar in water; and HFD + RF: high-fat diet adjusted with 25% Ramon flour + 5% sugar in water; AUC: area under curve.

**Figure 4 antioxidants-12-01957-f004:**
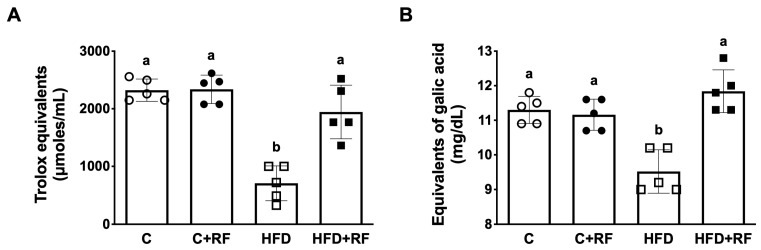
Antioxidant status in serum after consumption of Ramon flour. Antioxidant activity (**A**) and polyphenol concentration (**B**). Data are expressed as the mean ± SEM (*n* = 5). The data were analyzed using a one-way analysis of variance (ANOVA) test with a *post hoc* Tukey’s test. Differences between experimental groups are shown with letters, where a > b. Statistical differences are considered with a value of *p* < 0.05. C: control diet (AIN-93); C + RF: control diet adjusted with 25% Ramon flour; HFD: high-fat diet + 5% sugar in water; and HFD + RF: high-fat diet adjusted with 25% Ramon flour + 5% sugar in water.

**Figure 5 antioxidants-12-01957-f005:**
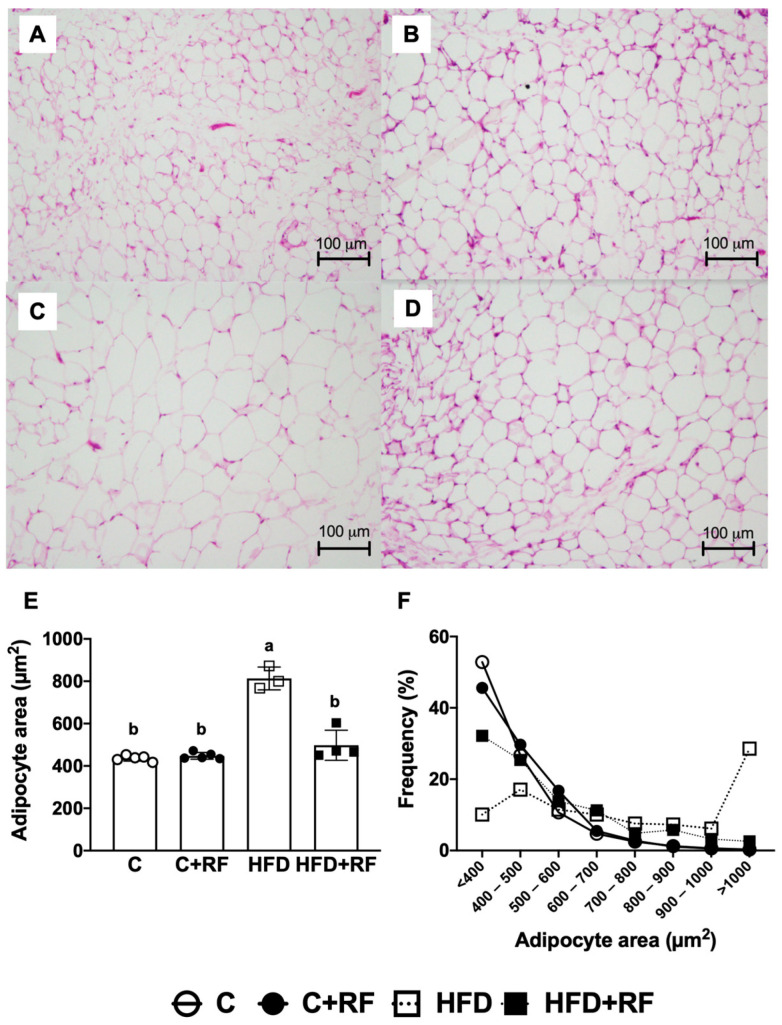
Histological analysis of white adipose tissue hematoxylin and eosin from groups after consumption of Ramon flour (10X magnification); Control (**A**), C + RF (**B**), HFD (**C**), and HFD + RF (**D**); adipocyte area (**E**) and adipocyte distribution size (**F**). Data are expressed as the mean ± SEM (*n* = 5). The data were analyzed using a one-way analysis of variance (ANOVA) test with a *post hoc* Tukey’s test. Differences between experimental groups are shown with letters, where a > b. Statistical differences are considered with a value of *p* < 0.05. C: control diet (AIN-93); C + RF: control diet adjusted with 25% Ramon flour; HFD: high-fat diet + 5% sugar in water; and HFD + RF: high-fat diet adjusted with 25% Ramon flour + 5% sugar in water.

**Figure 6 antioxidants-12-01957-f006:**
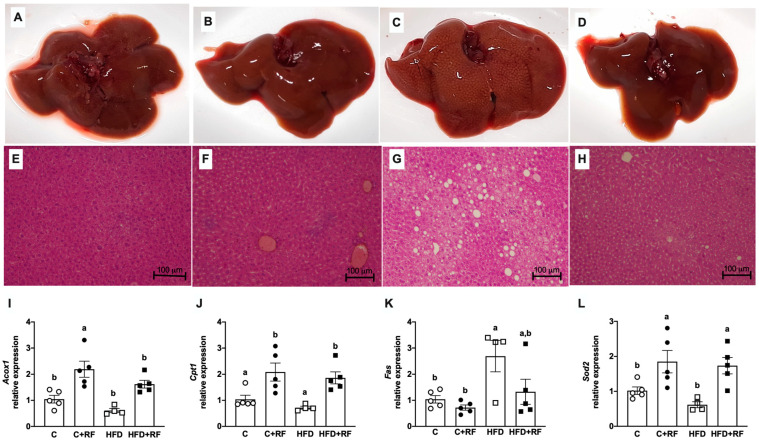
Macroscopic analysis of liver after consumption of Ramon flour (10X magnification) for Control (**A**), C + RF (**B**), HFD (**C**), and HFD + RF (**D**); hepatic hematoxylin and eosin histological results for Control (**E**), C + RF (**F**), HFD (**G**), and HFD + RF (**H**); hepatic expression of genes involved in lipogenesis, β-oxidation, and antioxidants (**I**–**L**). Data are expressed as the mean ± SEM (*n* = 5). The data were analyzed using a one-way analysis of variance (ANOVA) test with a *post hoc* Tukey’s test. Differences between experimental groups are shown with letters, where a > b. Statistical differences are considered with a value of *p* < 0.05. C: control diet (AIN-93); C + RF: control diet adjusted with 25% Ramon flour; HFD: high-fat diet + 5% sugar in water; and HFD + RF: high-fat diet adjusted with 25% Ramon flour + 5% sugar in water; *Acox1*: acyl-CoA oxidase 1 gene; *Cpt1*: carnitine palmitoyl transferase 1 gene; *Fas*: fatty acid synthase gene; *Sod2*: superoxide dismutase 2 gene.

**Figure 7 antioxidants-12-01957-f007:**
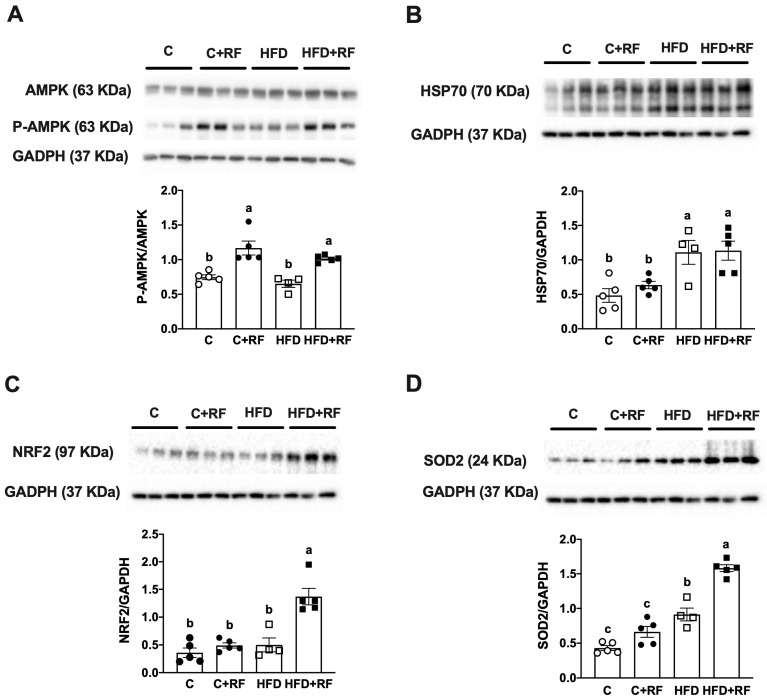
Effect of Ramon flour on hepatic protein abundance of AMPK and P-AMPK (**A**); HSP70 (**B**); NRF2 (**C**), and SOD2 (**D**) as assessed with GADPH used as the loading control. Data are expressed as the mean ± SEM (*n* = 5). The data were analyzed using a one-way analysis of variance (ANOVA) test with a *post hoc* Tukey’s test. Differences between experimental groups are shown with letters, where a > b > c. Statistical differences are considered with a value of *p* < 0.05. C: control diet (AIN-93); C + RF: control diet adjusted with 25% Ramon flour; HFD: high-fat diet + 5% sugar in water; and HFD + RF: high-fat diet adjusted with 25% Ramon flour + 5% sugar in water.

**Table 1 antioxidants-12-01957-t001:** Chemical analysis of macronutrients of experimental diets (g/100 g diet).

Component	Control	C + RF	HFD	HFD + RF
Carbohydrate	55.4 ± 0.49 ^a^	54.9 ± 0.19 ^a^	35.9 ± 0.29 ^b^	36.4 ± 0.19 ^b^
Protein	19.0 ± 0.52 ^b^	20.2 ± 0.24 ^b^	21 ± 0.23 ^a^	23 ± 0.24 ^a^
Lipid	7.18 ± 0.12 ^b^	7.9 ± 0.19 ^b^	30.3 ± 0.36 ^a^	28.7 ± 0.54 ^a^
Fiber	6.6 ± 0.22 ^a^	5.19 ± 0.17 ^b^	5.5 ± 0.27 ^b^	4.50 ± 0.03 ^c^
Mineral	4.59 ± 0.06 ^a^	4.22 ± 0.08 ^a^	4.46 ± 0.01 ^a^	4.66 ± 0.05 ^a^
Energy (Kcal/g)	3.75	3.81	5.11	5.05

Data are shown as mean ± SD (*n* = 3). The data were analyzed using one-way analysis of variance (ANOVA) test with a *post hoc* Tukey’s test. Differences between experimental groups are shown with letters, where a > b > c. Statistical differences are considered with a value of *p* < 0.05. C: control diet (AIN-93); C + RF: control + Ramon flour; HFD: high-fat diet + 5% sugar in water; HFD + RF: high-fat diet + Ramon flour + 5% sugar in water.

## Data Availability

The data presented in this study are available on request from the corresponding author.

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
