# Peer review of "Ramon Flour (Brosimum alicastrum Swartz) Ameliorates Hepatic Lipid Accumulation, Induction of AMPK Phosphorylation, and Expression of the Hepatic Antioxidant System in a High-Fat-Diet-Induced Obesity Mouse Model"

_antioxidants, 2023, doi:10.3390/antiox12111957_

Round 1

Reviewer 1 Report

The manuscript entitled Ramon flour (Brosimum alicastrum Swartz) ameliorates hepatic lipid accumulation and induced of AMPK phosphorylation and expression of the hepatic antioxidant system in a high-fat diet-induced obesity mouse model is an original study. The authors assessed the metabolic effects of Ramon flour consumption on a high-fat diet-induced obesity mouse model. Ramon flour has benefic, antioxidant effects on hepatic lipid accumulation and, therefore, the authors concluded that it might be used as a nutritional strategy to decrease metabolic alteration during obesity.

The results have shown that Ramon flour prevents biochemical alterations, lipid accumulation in liver and hypertrophy of white adipose tissue, induced by obesity, supporting the use of Ramon flour as a nutritional strategy to decrease unhealthy alterations in obesity.

The methods and results are well presented. Discussions are well conducted.

What are the limitations of this study?

This article is important because it is the first evidence showing the beneficial effects of Ramon flour consumption at the molecular level.

Author Response

Reviewer 1

The manuscript entitled Ramon flour (Brosimum alicastrum Swartz) ameliorates hepatic lipid accumulation and induced of AMPK phosphorylation and expression of the hepatic antioxidant system in a high-fat diet-induced obesity mouse model is an original study. The authors assessed the metabolic effects of Ramon flour consumption on a high-fat diet-induced obesity mouse model. Ramon flour has benefic, antioxidant effects on hepatic lipid accumulation and, therefore, the authors concluded that it might be used as a nutritional strategy to decrease metabolic alteration during obesity.

The results have shown that Ramon flour prevents biochemical alterations, lipid accumulation in liver and hypertrophy of white adipose tissue, induced by obesity, supporting the use of Ramon flour as a nutritional strategy to decrease unhealthy alterations in obesity.

1.The methods and results are well presented. Discussions are well conducted.

R1: We appreciated your comments about the manuscript.

2.What are the limitations of this study?

R2: We added the following paragraph about limitations in the discussion section (Page 13, lines 460-470):

The present study has some limitations. Firstly, no significant difference was observed in the body weight of the animals among the different groups. However, despite this, metabolic alterations associated with HFD consumption were observed. Another limitation is the absence of specific identification of bioactive compounds. As such, the benefits of RF may be attributed to the combination of various compounds, including fiber, vegetable protein, and unidentified bioactive compounds. Although total polyphenols in serum were determined after consumption of the diets, specific compounds were not identified. Moreover, both the control and HFD reference diets had a higher fiber content. Strikingly, despite this higher fiber content, the associated benefits were not observed in these groups. It´s worth noting that the benefits of RF were demonstrated at both the physiological and biochemical levels, particularly in the liver at the molecular level.

 3.This article is important because it is the first evidence showing the beneficial effects of Ramon flour consumption at the molecular level.

R3: Thank you for your opinion with respect to the present work.

Reviewer 2 Report

Manuscript is interesting, evaluating the effect of RF on general biochemical and morphological changes in a diet induced obesity animal model.

There are some changes/clarifications needed at this time.

Data from table 1 shows that HFD as well as HFD+RF are characterized by a significantly lower level of carbohydrate. This is strange since authors state that these 2 groups receive 5% sugar added in water. Also, the fibre content is very different between the 4 groups.

These differences should be carefully discussed in the manuscript since they could induce errors in the interpretation of data.

Authors state that 3.1. Addition of Ramon Fluor to the control or high-fat diets did not change their chemical composition – but some important differences are noticed regarding the fibre content upon adding RF

Authors tested the total polyphenols in blood samples; this evaluation should be also performed to characterize the RF samples. Especially since authors state “Few studies have reported the content and identification of polyphenols and their derivatives in 360 Ramon seed.”

The glucose level in C+RF group is higher compared to C group. Justification should be provided.

Figure 3A should be re-designed because there is overlapping, and it is difficult to read.

The study limitations should be included in the discussion section.

Author Response

Reviewer 2

Manuscript is interesting, evaluating the effect of RF on general biochemical and morphological changes in a diet induced obesity animal model.

There are some changes/clarifications needed at this time.

1.Data from table 1 shows that HFD as well as HFD+RF are characterized by a significantly lower level of carbohydrate. This is strange since authors state that these 2 groups receive 5% sugar added in water. These differences should be carefully discussed in the manuscript since they could induce errors in the interpretation of data.

R1: Regarding your comments, we considered the volume of water consumed by animals in HFD groups (we added this information to the methods section, page 2, lines 99-100). In fact, the data that showed energy intake per day in HFD groups considers the kilocalories ingested by diet and the sugar contained in the water (we added this information to the results section, page 4, lines 180-182). This ensures that the consumption between groups is not different (in kcal/d) and that the effects are due to the addition of Ramon flour. Thus, the adjustment of diet macronutrients allows us to suggest that the observed changes are associated with the specific presence of Ramon flour, since the percentage of the other macronutrients is equal in their respective comparison groups (HFD vs. HFD+Ramon flour or Control vs. Control+Ramon flour).

2.Authors state that 3.1. Addition of Ramon Fluor to the control or high-fat diets did not change their chemical composition – but some important differences are noticed regarding the fibre content upon adding RF.

R2: Despite the fact that the HFD and control diets contained more fiber, the benefits associated with the presence of fiber were not observed in these groups. This could be associated with the beneficial effects on fiber being between 10-20% in the diet (Cao Y, et l., 2011; Ogata M, et. al., 2017; Cheng W, et. Al., 2018). 

References

  • Cao Y, Gao X, Zhang W, Zhang G, Nguyen AK, Liu X, Jimenez F, Cox CS Jr, Townsend CM Jr, Ko TC. Dietary fiber enhances TGF-β signaling and growth inhibition in the gut. Am J Physiol Gastrointest Liver Physiol. 2011 Jul;301(1):G156-64. doi: 10.1152/ajpgi.00362.2010.
  • Ogata M, Ogita T, Tari H, Arakawa T, Suzuki T. Supplemental psyllium fibre regulates the intestinal barrier and inflammation in normal and colitic mice. Br J Nutr. 2017 Nov;118(9):661-672. doi: 10.1017/S0007114517002586.
  • Cheng W, Lu J, Lin W, Wei X, Li H, Zhao X, Jiang A, Yuan J. Effects of a galacto-oligosaccharide-rich diet on fecal microbiota and metabolite profiles in mice. Food Funct. 2018 Mar 1;9(3):1612-1620. doi: 10.1039/c7fo01720k.

 3.Authors tested the total polyphenols in blood samples; this evaluation should be also performed to characterize the RF samples. Especially since authors state “Few studies have reported the content and identification of polyphenols and their derivatives in 360 Ramon seed.”

 R3: As suggested, the total polyphenol content of Ramon flour (52.9 ± 3.78 GAE/g) present in the diets is added (page 7, lines 239-240), and it is shown that this content is similar to previous report (65.8 ± 2.26 mg GAE/g) (Subiria-Cueto R et. al., 2019)(we added this information also en discussion section, page 12 lines 372-373).

References 

  • Subiria-Cueto R, Larqué-Saavedra A, Reyes-Vega ML, de la Rosa LA, Santana-Contreras LE, Gaytán-Martínez M, Vázquez-Flores AA, Rodrigo-García J, Corral-Avitia AY, Núñez-Gastélum JA, Martínez-Ruiz NR. Brosimum alicastrum Sw. (Ramón): An Alternative to Improve the Nutritional Properties and Functional Potential of the Wheat Flour Tortilla. Foods. 2019 Nov 24;8(12):613. doi: 10.3390/foods8120613.

4.The glucose level in C+RF group is higher compared to C group. Justification should be provided.

R4: Although an increase in glucose levels was observed in the C+RF group, it does not show a statistically significant difference from the control group. The levels of both groups remained in normal ranges (<150 mg/dL) (Kobori M, et al., 2012; Valdes M, et al., 2019; Albawardi A, et al., 2023).

 References 

  • Kobori M, Masumoto S, Akimoto Y, Oike H. Phloridzin reduces blood glucose levels and alters hepatic gene expression in normal BALB/c mice. Food Chem Toxicol. 2012 Jul;50(7):2547-53. doi: 1016/j.fct.2012.04.017.
  • Valdes M, Calzada F, Mendieta-Wejebe J. Structure-Activity Relationship Study of Acyclic Terpenes in Blood Glucose Levels: Potential α-Glucosidase and Sodium Glucose Cotransporter (SGLT-1) Inhibitors. Molecules. 2019 Nov 6;24(22):4020. doi: 10.3390/molecules24224020.
  • Albawardi A, Saraswathiamma D, Sharma C, Elomami A, Souid AK, Almarzooqi S. Effect of Sirolimus/Metformin Co-Treatment on Hyperglycemia and Cellular Respiration in BALB/c Mice. Int J Mol Sci. 2023 Jan 8;24(2):1223. doi: 10.3390/ijms24021223.

5.Figure 3A should be re-designed because there is overlapping, and it is difficult to read.

R5: As you suggested, we modified the figure and the legend to improve their understanding.

 6.The study limitations should be included in the discussion section.

R6: We added the following paragraph about limitations in the discussion section (Page 13, lines 460-470):

The present study has some limitations. Firstly, no significant difference was observed in the body weight of the animals among the different groups. However, despite this, metabolic alterations associated with HFD consumption were observed. Another limitation is the absence of specific identification of bioactive compounds. As such, the benefits of RF may be attributed to the combination of various compounds, including fiber, vegetable protein, and unidentified bioactive compounds. Although total polyphenols in serum were determined after consumption of the diets, specific compounds were not identified. Moreover, both the control and HFD reference diets had a higher fiber content. Strikingly, despite this higher fiber content, the associated benefits were not observed in these groups. It´s worth noting that the benefits of RF were demonstrated at both the physiological and biochemical levels, particularly in the liver at the molecular level.

Round 2

Reviewer 2 Report

Authors improved the manuscript based on the comments received